# Comprehensive Evaluation of Nutritional Quality Diversity in Cottonseeds from 259 Upland Cotton Germplasms

**DOI:** 10.3390/foods14162895

**Published:** 2025-08-20

**Authors:** Yiwen Huang, Chengyu Li, Shouyang Fu, Yuzhen Wu, Dayun Zhou, Longyu Huang, Jun Peng, Meng Kuang

**Affiliations:** 1Zhengzhou Research Base, State Key Laboratory of Cotton Bio-Breeding and Integrated Utilization, School of Agricultural Sciences, Zhengzhou University, Zhengzhou 450001, China; huangyiwen@caas.cn (Y.H.); 13783886891@163.com (C.L.); 18438616378@163.com (S.F.); 15959263920@163.com (Y.W.); zhoudayun@caas.cn (D.Z.); huanglongyu1@163.com (L.H.); 2Western Agricultural Research Center, Chinese Academy of Agricultural Sciences, Changji 831100, China; 3Sanya National Nan Fan Research Institute, Chinese Academy of Agricultural Sciences, Sanya 572024, China

**Keywords:** *Gossypium hirsutum* L., cottonseed, nutritional quality, fiber quality, comprehensive evaluation

## Abstract

Cottonseeds, rich in high-quality protein and fatty acids, represent a vital plant-derived feedstuff and edible oil resource. To systematically investigate genetic variation patterns in nutritional quality and screen superior germplasm, this study analyzed 26 nutritional quality traits and 8 fiber traits across 259 upland cotton (*Gossypium hirsutum* L.) accessions using multivariate statistical approaches. Results revealed significant genetic diversity in cottonseed nutritional profiles, with coefficients of variation ranging from 3.42% to 26.37%. Moreover, with advancements in breeding periods, the contents of protein, amino acids, and the proportion of unsaturated fatty acids (UFAs) increased, while oil content and C16:0 levels decreased. Correlation analyses identified significant positive associations (*p* < 0.05) between proteins, amino acids, UFAs, and most fiber traits, except for seed index (SI), fiber micronaire (FM), and fiber elongation (FE). Through a principal component analysis–fuzzy membership function (PCA-FMF) model, 13 elite accessions (F > 0.75) with high protein content, high UFA proportion, and excellent fiber quality were identified. These findings provide both data-driven foundations and practical germplasm resources for value-added utilization of cottonseed and coordinated breeding for dual-quality traits of nutrition and fiber.

## 1. Introduction

Cotton (*Gossypium* spp.), as one of the world’s most important economic crops, serves as the cornerstone of the global textile industry [1]. Cottonseed, the primary byproduct of cotton production, yields approximately 1.65 kg of cottonseed per 1 kg of cotton fiber produced [2]. Rich in high-quality protein and oil, cottonseed ranks as the second-largest plant protein source and fifth-largest oilseed crop globally [3,4]. Cottonseed protein boasts a balanced amino acid profile, containing all eight essential amino acids (EAAs) required by humans, and exhibits multiple physiological functions including antioxidant, antihypertensive, and immune-enhancing properties [5]. Defatted cottonseed meal, with a crude protein content of 43.5–47.0%, is a widely utilized plant protein source, second only to soybean meal in nutritional value, and serves as a premium protein feed ingredient [6]. Furthermore, cottonseed protein concentrate produced through low-temperature extraction, dephenolization, and moderate desugarization processes, which is free from allergenic antigens, significantly enhances the economic value of cottonseed protein, and can effectively replace fish meal and soybean protein [7,8]. Moreover, cottonseed protein can serve as a high-quality emulsifier after sulfonation modification [9]. Cottonseed oil, abundant in UFAs and vitamin E, demonstrates multiple biological functions including antioxidant, anti-inflammatory, anticancer, and wound-healing properties, and its regular consumption contributes to cardiovascular health maintenance [10]. It is favored for its mild flavor, which does not overpower the natural taste of food, and its high smoke point makes it superior to other oils and fats for frying applications [11]. Additionally, cottonseed oil finds applications in producing biodiesel, lubricants, and hydraulic oils [12]. With rapid global population growth and rising living standards, the demand for protein and oil resources continues to escalate [11], necessitating the development of novel protein and oil sources. Therefore, enhancing the comprehensive utilization value of cottonseed and breeding cotton varieties with superior nutritional quality holds significant strategic importance.

Cottonseed nutritional quality exhibits significant genetic variation both between and within species. Hinze et al. conducted a comprehensive analysis of 2256 samples from 33 cotton species, revealing extensive variation ranges in protein and oil content within upland cotton (*Gossypium hirsutum* L.) and sea island cotton (*Gossypium barbadense* L.) [13]. As the most widely cultivated cotton species in the world, upland cotton accounts for approximately 90% of global cotton production [14]. Furthermore, Hu et al. [4] quantified protein and oil content in 318 upland cotton accessions, revealing variation ranges of 34.04–45.68% for protein content and 27.19–39.89% for oil content. Additionally, gas chromatography analysis of five major fatty acids in 196 cultivars confirmed highly significant genotypic variation (*p* < 0.01), with an average variation index of 4.55% across fatty acid components [15]. These consistent findings highlight the rich genetic diversity in cottonseed nutritional quality, providing a crucial germplasm foundation for genetic improvement and value-added utilization [16].

Cottonseed protein and oil are present in the ovule, while fibers develop from trichomes, which are single-celled protuberances of the ovule epidermis [17]. A potential correlation may exist between the cottonseed nutritional quality and the fiber trait. Campbell et al. conducted measurements of traits in 82 American cotton germplasms across multiple environments, finding a positive correlation between cottonseed protein content and fiber traits such as lint yield, lint percentage (LP), fiber micronaire (FM), while the cottonseed oil content was positively correlated with seed index (SI) and fiber strength (FS) [18]. Similar results were reported by Hu et al., who found a positive correlation (*r* = 0.65, *p* < 0.05) between cottonseed protein content and LP [4]. Conversely, studies have reported a negative correlation between LP and cottonseed protein content, as well as a negative correlation between cottonseed oil content and SI [19], along with a positive correlation between cottonseed oil content and traits such as LP, fiber length (FL), fiber uniformity (FU), and fiber strength (FS) [20]. Discrepancies exist in the results reported by different studies. Thus, the relationship between these two traits remains to be further elucidated. As the two main products of cotton production, the synergistic improvement of fiber and cottonseed is crucial for enhancing the overall economic benefits of cotton.

Proteins and oils are not only essential nutritional components required by humans and animals but also important strategic resources that promote economic growth and support sustainable development [21]. Cottonseed, as a high-quality source of plant protein and oil, holds potential to alleviate global food security issues through cultivating new varieties with excellent nutritional quality and improving their comprehensive utilization rates. However, the high-value utilization of cottonseed currently faces two major bottlenecks: (1) a lack of systematic foundational data support, (2) insufficient screening of high-quality germplasm resources, hindering efforts to meet the demands of targeted breeding and comprehensive utilization. Although preliminary evaluations of the nutritional qualities exist, many studies focus primarily on protein and oil content. This study utilizes 259 upland cotton germplasms as experimental materials to systematically evaluate 26 nutritional quality traits (including protein, oil, 16 amino acids, and 8 fatty acids) and 8 fiber quality traits (3) fiber yield traits and 5 fiber quality traits)). By employing multivariate statistical analysis methods, we comprehensively investigate the variation characteristics and distribution patterns of cottonseed nutritional quality. Furthermore, through correlation analysis between nutritional components and fiber quality traits, combined with a principal component analysis–fuzzy membership function (PCA-FMF) comprehensive evaluation model, we aim to identify germplasm resources with superior comprehensive traits. These findings provide critical data support and a theoretical foundation for the high-value utilization of cottonseed and the development of new cotton varieties with enhanced nutritional and fiber quality.

## 2. Materials and Methods

### 2.1. Materials and Field Experiment Design

A total of 259 upland cotton accessions were selected for the experiment (Appendix A), including 40 accessions from the Yangtze River Basin (YZR), 95 accessions from the Yellow River Basin (YER), 28 accessions from the northern extra-early-maturing cotton region (NER), 58 accessions from the northwest inland region (NIR), 15 landraces (LAN), and 23 foreign accessions (FOR). All accessions were provided by the Cotton Research Institute, Chinese Academy of Agricultural Sciences. In the 2023 cotton-growing season, they were planted at the Anyang experimental station, Henan (36°10′ N, 114°38′ E). Each accession was planted in a single 5 m long row with a row spacing of 0.8 m and an isolation path of 1 m between plots. Field management followed local standard practices.

### 2.2. Sample Pretreatment

At maturity, 50 normally developed cotton bolls were collected from the middle section of plants for each accession. After naturally air-drying, the samples were processed using a ginning machine to separate the cotton fibers from the cotton seeds. The seeds were then manually delinted to obtain kernels. An IKA mill (IKA-Werke GmbH & Co. KG, Staufen, Germany) was used to grind the cottonseed kernels to a fineness of 60 mesh (250 μm). All samples were dried in a constant-temperature drying oven at 45 °C until they reached constant weight (difference between consecutive weighing ≤0.1%) and subsequently were then stored in a desiccator for future use.

### 2.3. Determination of Cottonseed Protein and Oil

In this study, the Kjeldahl method and Soxhlet extraction method were used to determine the protein content and oil content of cottonseeds from 259 cotton accessions. Protein content was determined as follows: precisely weigh 0.5000 g of cotton kernel powder and add a mixed catalyst (K_2_SO_4_: CuSO_4_·5H_2_O = 7:0.8) along with 10 mL of concentrated sulfuric acid. Digest the mixture at 420 °C for 40 min until the solution turns a transparent blue-green. Transfer the digested solution to a Kjeldahl nitrogen analyzer (FOSS Analytical A/S, Hillerød, Denmark), alkalize it with 40% NaOH, and distill the solution. Absorb the distillate using 2% boric acid and titrate with standard hydrochloric acid until a pink endpoint is reached [22]. Record the titration volume. Include a blank control for each batch of samples. The nitrogen-to-protein conversion factor used is 6.25.

Oil content was determined as follows: First, dry the cotton kernel powder at 105 °C until a constant weight is achieved. Accurately weigh 2 g of the dried sample, wrap it in pre-dried and weighed filter paper, and place it in a Soxhlet extractor. Use petroleum ether (boiling range: 30–60 °C) as the extraction solvent and perform the extraction in a 50 °C water bath for 10 h. After extraction, dry the filter paper package at 105 °C until a constant weight is achieved. Calculate the oil content using the mass difference method [22].

### 2.4. Determination of Cottonseed Amino Acids

Amino acid contents were determined using post-column derivatization ion exchange chromatography with ninhydrin on a Hitachi L-8900 amino acid analyzer (Hitachi, Ltd., Tokyo, Japan): Weigh 5.000 g (±0.001 g) of homogenized cotton kernel powder, mix with 6 mol/L HCl, and hydrolyze at 110 °C for 24 h. Filter the hydrolysate, dilute to 50 mL, and analyze. Instrument conditions: column temperature 57 °C, reactor temperature 135 °C, buffer flow rate 0.4 mL/min, and ninhydrin flow rate 0.3 mL/min. Inject the mixed amino acid standard and sample solutions. Amino acid concentrations were quantified using the external standard method. Each sample was analyzed in duplicate, and the average value was used for calculations. The criterion for data acceptance was set as the absolute difference between the two duplicate measurements not exceeding 12% of their arithmetic mean under repeatability conditions. Amino acid standards purchased from Wako Pure Chemical Industries, Ltd. (Osaka, Japan) were used for calibration. Amino acid content was calculated based on dry weight. Analyze a standard sample every 10 samples for quality control [23].

### 2.5. Determination of Cottonseed Fatty Acids

In this study, the fatty acid composition of 259 cotton accessions was analyzed using an Agilent 6890N gas chromatograph (Agilent Technologies, Inc., Santa Clara, CA, USA). The procedure was as follows: 0.1000 g of cotton kernel powder was hydrolyzed with 5 mL of 6 mol/L HCl in an 80 °C water bath for 40 min. After hydrolysis, 10 mL of 95% ethanol was added, and the sample was extracted three times with an ether–petroleum ether (1:1) mixture. Combine the organic phases, concentrate under nitrogen at 40 °C, and dry to obtain the fat extract. Transesterify the extract with 8 mL of 2% NaOH–methanol solution at 80 ± 1 °C for 30 min to produce fatty acid methyl esters (FAMEs). FAMEs were analyzed using a DB-23 capillary column (60 m × 0.25 mm × 0.25 μm) with the following conditions: injector temperature 250 °C, detector temperature 280 °C, and column temperature programmed from 100 °C to 240 °C at 4 °C/min, holding for 15 min. High-purity nitrogen was used as the carrier gas at 1.0 mL/min, with a split ratio of 50:1. Each sample was also analyzed in duplicate, with the acceptance criterion being that the absolute difference between duplicates did not exceed 10% of their arithmetic mean. FAME standards from Aladdin Technology Co., Ltd. (Tianjin, China) were used for identification and quantification [24]. Include a quality control sample every 20 samples [25].

### 2.6. Determination of Yield and Fiber Quality Traits of Upland Cotton

To comprehensively evaluate fiber traits, the fiber characteristics of 259 cotton germplasms were measured prior to analyzing cottonseed nutritional quality. The following traits were assessed: First, the weight of 20 bolls weight (BW), LP, and SI were determined using an electronic balance. Additionally, 20 g fiber samples from each accession were randomly selected and sent to the Cotton Quality Supervision and Inspection Center of the Ministry of Agriculture and Rural Affairs (Anyang, China) for professional testing. The tested fiber quality traits included FL, FU, FS, FM, and fiber elongation (FE). Each sample was analyzed in triplicate, and the average value was used to minimize experimental error.

### 2.7. Comprehensive Evaluation of Upland Cotton Quality

To comprehensively evaluate the 259 upland cotton accessions, a PCA-FMF model was employed to calculate comprehensive scores (F value) based on 26 cottonseed nutritional quality traits and 8 fiber quality traits. First, principal component analysis (PCA) was used to reduce the dimensionality of the original trait data. Extract principal components (PCs) with eigenvalues greater than 1. Calculate the contribution rate (CRi) of each PC and its weight coefficient (Wi): Wi = CRi/ΣCRi. Next, the fuzzy membership function (FMF) method was applied to normalize the PCs. The principal component scores (Xi) for each accession were normalized as follows: U (Xi) = (Xi − Xmin)/(Xmax − Xmin), where Xmin and Xmax are the minimum and maximum scores for that PC, respectively. The F value for each accession was calculated using the formula: F = Σ(Ui × Wi) [26]. This study selects the top 5% of F values as superior germplasm.

### 2.8. Data Statistical Analysis

Basic statistical analysis was conducted using IBM SPSS Statistics 25 (IBM, Armonk, NY, USA). Describe the distribution characteristics of all traits using descriptive statistics (mean ± standard deviation). A one-way analysis of variance (ANOVA) combined with Tukey’s post hoc test was employed to determine the significance of differences between groups (*p* < 0.05). Calculate the Pearson correlation coefficient matrix to explore relationships among cottonseed nutritional qualities and fiber traits. Perform PCA using the factoextra package in R Studio (version 4.2.1). Basic visualizations, including histograms and box plots, were created using OriginPro 2022 (Originlab Corporation, Northampton, MA, USA).

## 3. Results

### 3.1. Descriptive Analysis of Nutritional Quality of Upland Cotton Seeds

In the 259 cotton germplasms, cottonseed protein, oil, 16 amino acids, and 8 fatty acids were identified (Appendix A). The results indicate that the primary components of cottonseed are protein (Figure 1A) and oil (Figure 1B), collectively accounting for 76.46% of total cottonseed kernel dry weight. A significant negative correlation was observed between cottonseed protein content and oil content, with a correlation coefficient of *r* = −0.947 (*p* < 0.01) (Figure 1C). Protein is the most abundant component in cottonseed, with an average content of 44.00% and a variation range of 35.32–53.48%. Among the 16 detected amino acids, 8 are EAAs, and 8 are non-essential amino acids (NAAs) (Figure 1D). Glu, Arg, and Asp are the three most abundant amino acids, accounting for 8.19%, 5.02%, and 3.60% of the cottonseed kernel, respectively, with variation ranges of 6.32–9.68%, 3.54–6.60%, and 2.78–4.48%. Lys, a common limiting amino acid in plant-based feeds, has a content range of 1.48% to 2.12%. The coefficient of variation (CV) for the 16 amino acids ranges from 7.79% to 13.07%, with Asp, Tyr, Phe, His, Arg, and Pro showing CVs exceeding 10%. Oil is the second major component of cottonseed, with an average content of 32.46% and a variation range of 25.35–41.16%. Among the eight fatty acids (Figure 1E), saturated fatty acids (SFAs: C14:0, C16:0, C18:0, C20:0) averaged 26.73%, while unsaturated fatty acids (UFAs: C16:1, C18:1, C18:2, C18:3) averaged 73.12% (Figure 1F). C18:2 is the predominant fatty acid in cottonseeds, accounting for 55.92% of the total fatty acids, with a variation range of 49.20–69.10%. Next in abundance are C16:1 (22.88%) and C18:1 (16.41%). The CV for the eight fatty acids ranges from 3.42% to 26.37%, with C18:2 and C14:0 showing higher CVs of 26.37% and 13.74%, respectively. Overall, the nutritional quality traits of cottonseed exhibit rich genetic variation across the 259 germplasms.

### 3.2. Analysis of Differences in Nutritional Quality of Cottonseeds Among Different Geographical Sources and Breeding Periods

To investigate the distribution and variation in cottonseed nutritional quality across different cotton-growing regions and breeding periods. Accessions were classified by geographical origin (six groups: YZR, LAN, FOR, YER, NER, NIR) and breeding period (four groups: foreign introduction, 1953–2000, 2001–2010, 2011–2022). By comparing protein, oil, amino acid content, and fatty acid composition among these groups, significant differences were observed among upland cotton accessions from different geographical origins. Among 26 traits, 23 exhibited highly significant differences (*p* < 0.05); C14:0, C18:0, and C16:1 showed no significant variation (Appendix A). FOR accessions exhibited the highest oil content at 35.58% (Figure 2A) and SFA proportion (27.15%) compared to other regions. However, these accessions had lower protein content, total amino acid content and C18:2 proportion. NIR accessions had the highest protein (45.65%) and all 16 amino acids. Although their oil content was lowest (30.75%), their C18:2 proportion was highest (56.34%). Overall, the NIR and YER accessions exhibited higher protein and amino acid content and C18:2 proportion, while FOR, LAN, YZR, and NER accessions had higher oil content (Figure 2C), which is a noteworthy finding. Significant differences were also found among breeding periods (Appendix A). Among 26 traits, 23 differed significantly (*p* < 0.05); C14:0, C18:1, and C18:3 showed no significant differences. (*p* < 0.05). Cottonseed nutritional quality changed systematically with breeding advancement: protein, amino acids, and UFA proportion increased, while oil content and C16:0 proportion decreased (Figure 2B,D). The proportion of SFA also decreased, whereas the proportion of UFA increased. Varieties bred after 2000 had significantly higher protein and amino acid content compared to earlier varieties, but their oil content and proportion of C16:0 were significantly lower (Figure 2D).

### 3.3. Phenotypic Variation in Fiber Quality

Fiber is the primary product of cotton production. To clarify the basic variation in fiber yield and fiber quality traits, three fiber yield traits (LP, BW, SI) and five fiber quality traits (FL, FS, FU, FE, FM) were measured (Appendix A). All traits exhibited normal distributions. CVs ranged from 1.46% (FU) to 13.34% (BW); four traits had CV > 10%. Yield traits showed greater variation (all CV > 10%) than quality traits, with BW ranging from 71.41 g to 163.63 g. Among quality traits, only FM had CV > 10% (Appendix A).

### 3.4. Correlation Analysis of Cottonseed Nutritional Quality and Fiber Traits

To explore the interrelationships between cottonseed nutritional quality and fiber traits, correlation analysis among cottonseed nutritional quality, and fiber traits across 259 accessions revealed significant intra- and inter-group correlations (Figure 3). Among cottonseed nutritional quality traits, protein content was positively correlated with all 16 amino acids, with correlation coefficients ranging from 0.57 to 0.85 (*p* < 0.01). Interestingly, protein content also exhibited weak positive correlations with the proportion of C18:2 (*r* = 0.26) and C18:3 (*r* = 0.25), respectively (*p* < 0.01). Due to the positive correlation between protein and amino acids, oil content showed significant negative correlations with all amino acids. In the relationships between cottonseed nutritional quality and fiber traits, protein content was positively correlated with five fiber traits (FL, FS, FU, LP, BW; *r* = 0.13–0.47), The highest correlation was observed with LP (*r* = 0.47). Oil content was positively correlated with SI, with a correlation coefficient of 0.44. As the predominant fatty acid, C18:2 demonstrated positive correlations with BW, LP, FL, FS, and FU. LP exhibited significant positive correlations with all 16 amino acids, with correlation coefficients ranging from 0.19 to 0.53. Among them, His showed the highest correlation with LP. Fiber traits such as LP, BW, FL, and FS have been key targets in cotton breeding programs. Our study found significant positive correlations between these traits and nutritional parameters (protein content, amino acid composition, C18:2 proportion). These findings support coordinated improvement of nutritional and fiber quality, promising simultaneous enhancements in fiber yield/quality and seed nutritional value—potentially boosting overall cotton production value.

### 3.5. Comprehensive Evaluation of Cotton Germplasm Resources

To identify elite cotton germplasm, we implemented a PCA-FMF integrated model for comprehensive evaluation and cluster analysis of all accessions. PCA revealed that PC1 and PC2 accounted for 57.07% of the total variation. PC1 was positively correlated with protein content and all amino acid content but negatively correlated with oil content and seed index. PC2 was primarily influenced by C18:2, C16:0, C20:0, FL, and FS (Figure 4A, Appendix A). Using the PCA-FMF model, a comprehensive evaluation was conducted on 259 cotton accessions; the F value ranged from 0.185 (Shaanmian 4) to 0.878 (Xinluzao 29), with an average of 0.539, and 58.59% of accessions exceeded the average F value. (Appendix A). The F value was significantly positively correlated with protein, all amino acids, C18:2, C18:3, BW, LP, FL, FS, FU, and FM, while it was significantly negatively correlated with oil content, relative content of C18:1, and SI (Appendix A). Further cluster analysis of the 34 traits divided the 259 accessions into four groups, with 24, 49, 95, and 91 accessions in each group, respectively (Figure 4B). Among the four groups, all 30 traits except for C14:0, C20:0, FM, and FE showed significant differences (*p* < 0.05) (Appendix A). Group 1 and group 2 were similar, consisting mainly of accessions from the YZR, FOR, and LAN with older breeding periods. These groups had higher oil content, C16:0, and SI compared to the other groups but exhibited poorer protein, amino acid, and fiber yield and quality traits. The varieties in groups 3 and 4 showed similar phenotypes, both having higher levels of protein, all amino acids, C18:1, C18:2, and BW. Compared with group 3, accessions of group 4 had optimal LP, FL, FS, and FU. Overall, group 4 demonstrated superior comprehensive traits (Figure 4C). To screen for germplasm resources with both excellent cottonseed nutritional quality and fiber quality, selecting the top 5% based on F value identified 13 elite accessions (e.g., Xinluzao29, Xinluzao28, Xinluzao48, Chuang0712, Xinluzao16, SkgZhong156, Xinluzao24, Shiyuan345, Xinluzhong2, Zhongmian113, Simian3, Jinghuamian116, Zhongjiangmian8) with high protein and amino acid contents, a favorable proportion of UFAs, and excellent fiber quality and yield (Figure 4D).

## 4. Discussion

### 4.1. Cottonseed as an Important Source of Protein and Oil

Proteins and oils are indispensable resources for humans and the livestock industry. As society and the livestock sector develop rapidly, coupled with constraints on soybean production, protein and oilseed resources are facing shortages [27]. Cotton is not only the main source of natural fiber, but its cottonseeds also harbor abundant nutrients [22]. Enhancing the comprehensive utilization rate of cottonseeds can help mitigate food security problems. In this study, a systematic evaluation was carried out on the protein and oil content, as well as the amino acid and fatty acid components of cottonseeds from 259 upland cotton accessions. The results showed that the dominant components in cottonseeds were proteins and oils. The protein content accounted for 44% of the weight of cottonseed kernels. Compared with soybeans, the total protein content of cottonseeds was equivalent or slightly higher [28]. The composition and proportion of amino acids were the key factors affecting its nutritional value and applications. The amino acid composition of cottonseed protein was relatively balanced. The contents of glutamic acid and arginine were significantly higher than those in soybean protein; these amino acids enhance the umami taste and nutritional value of foods, making cottonseed protein an ideal candidate for developing flavor-enhancing protein additives [29]. In contrast, the content of lysine was relatively low [30]. Through deep-processing methods such as de-oiling and enzymatic hydrolysis, cottonseed meal can be transformed into a high-concentration cottonseed protein concentrate, which has higher economic value [7]. In terms of oils, the average oil content of upland cotton was 32.46%, slightly higher than that of soybeans, and the content of unsaturated fatty acids was similar. The content of linoleic acid in cottonseed oil was relatively high [31]. Compared with rapeseed oil, cottonseed oil contained almost no erucic acid, while the content of oleic acid was higher than that in cottonseed oil, but the content of linoleic acid was lower [32]. Traditional research suggested that the toxic substance gossypol in cottonseeds limited its effective utilization in feed and edible oil [33]. However, modern dephenolization processes are quite mature and can effectively remove excessive gossypol [34]. Wang et al. [35] also showed that low concentrations of gossypol have broad-spectrum resistance to animal viruses. Although cottonseed is a by-product of the cotton production process, its production level is also high. Cottonseed yield accounts for 60% of the total cotton production. According to data, from 2019 to 2022, approximately 166.29 million tons of cottonseeds were produced globally (https://www.fao.org/faostat/zh/#data/FBS, accessed on 10 March 2025). All these findings indicate that cottonseed, as a potential resource for proteins and oils, has good development prospects.

### 4.2. Wide Variation in Nutritional Quality of Upland Cotton Seeds

Upland cotton is currently the most widely cultivated cotton species, accounting for 99% of China’s cotton planting area. Despite its relatively narrow genetic background compared with other wild and semi-wild cotton varieties [36], it exhibits extensive variation in nutritional quality. Compared with other diploid cotton species, it boasts the highest oil content and the second highest protein content [13]. Although the reported variation ranges of cottonseed protein and oil contents differ across studies, the general ranges of cottonseed protein and oil contents are approximately 31.92–55.13% and 22.91–41.98%, respectively [22,37,38], which are basically consistent with the results of our study. These disparities can be attributed to multiple factors, including sample types [39], determination methods [18], and calculation approaches. In certain studies, protein and oil contents were computed relative to the weight of cottonseeds with shells [40]. This practice led to comparatively lower measured values. Nevertheless, the reported ranges for protein and oil contents remained at 15.26% and 14.66%, respectively [13]. In this study, we not only examined the protein and oil contents of upland cotton germplasms but also conducted a detailed evaluation and analysis of the amino acid and fatty acid components. The average variation index of the 26 nutritional quality component indicators was 10%, and the coefficient of variation of eight indicators exceeded 10%, indicating more intense variation in these components. This shows that there is extensive variation in the nutritional quality traits of cottonseeds in upland cotton, which can provide germplasms and parents for different industrial demands and breeding. Through correlation analysis, we found that there were significant positive correlations among the various amino acids in cottonseeds. Although the correlation of methionine was slightly weaker, it still exceeded 0.6. This phenomenon has also been observed in soybeans, peanuts, and avocados [41,42,43]. This positive correlation phenomenon may be linked to the amino acid biosynthesis pathway in seeds. Amino acids are mainly synthesized from the intermediate products of the tricarboxylic acid cycle [44]. α-ketoglutarate can be converted into glutamic acid, and the amino group of glutamic acid can generate different amino acids under the action of various transaminases [45]. This process is regulated by glutamine synthetase and glutamate synthase [46]. The relatively low correlation between methionine and other amino acids may be mainly due to its status as a sulfur-containing amino acid [47]. C16:0, C18:1, and C18:2 are the three main fatty acids in cottonseed oil, accounting for 95.21% of the total fatty acids. C18:2 is the most abundant fatty acid, with a content of 55.92%, which is consistent with the reports of Hu et al. [4] and Zhuang et al. [22]. The relationships among fatty acids are intricate. Overall, there is a significant negative correlation between saturated and unsaturated fatty acids, and C18:1 and C18:2 show a significant negative correlation. This is because under the action of stearoyl-acyl-carrier-protein desaturase, saturated fatty acids are dehydrogenated to form unsaturated fatty acids [48], and the biosynthesis of linoleic acid takes oleic acid as a substrate and is formed under the action of fatty acid desaturase [49], creating a competitive relationship.

### 4.3. Influence of Breeding Process on Cottonseed Nutritional Quality

Long-term natural variation and artificial selection have led to differences in the adaptability of cotton in different ecological environments [50]. Understanding the changes in key agronomic traits among different regions is crucial for the cultivation of new varieties. Our analysis revealed significant geographical and temporal differences in cottonseed nutritional quality, closely tied to Chinese cotton-breeding history. The breeding history of Chinese cotton has gone through stages from introducing foreign accessions to cultivating early-stage accessions. These early-stage accessions were mainly planted in the YZR and the YER [36]. Subsequently, the cotton planting area gradually shifted from southern China to northwestern China, and the varieties in the northwestern inland region developed rapidly [50]. Early accessions (foreign, landraces, early cultivars) had higher oil but lower protein/amino acids. Recent NIR varieties showed increased protein and C18:2 but decreased oil and C16:0. There is an association between the clustering results, geographical origin, and breeding trends; there are geographical distribution differences among different subgroups. This shift may relate to changing cultivation patterns, potentially involving fatty acids degradation for water production under drought stress in arid regions [51].

### 4.4. Simultaneous Improvement of Cottonseed Nutritional Quality and Fiber Quality

Fibers and cottonseeds are the two main products of the cotton production process. Fibers develop from the single cell trichome protrusions on the ovule, and there is an inseparable relationship between them. We found protein positively correlated with LP and fiber quality traits, while the oil content was significantly positively correlated with the SI, which is consistent with previous reports [4,52]. Additionally, this study also found that C18:2 was positively correlated with LP and fiber quality traits excluding FE. Interestingly, although early cotton breeding primarily focused on fiber yield, quality, and disease resistance [1], neglecting cottonseed, our study found that cottonseed nutritional quality has co-evolved with breeding trends and converged with fiber quality traits. This phenomenon is likely driven by genes with pleiotropic effects or linked genetic loci that co-regulate. A genome-wide association study by Hu et al. [4] identified 54 loci associated with cottonseed protein, oil content, and fatty acid composition, among which 35 loci overlapped with those controlling fiber quality and yield traits. Specifically, the locus A02:79231003, which was associated with seed oil content, coincided with five signals linked to SI and one signal related to LP. Furthermore, Chapman et al. [52] has shown that ectopic overexpression of *BnFAD2* in cotton increases seed C18:2 contents while enhancing LP. The redirected carbon flux, driven by *FAD2* expression, prioritizes fiber development over oil biosynthesis in seeds [52]. Recent research has also found that *GhD53* inhibits fiber elongation by suppressing the transcription of the *GhFAD3* gene, which controls the biosynthesis of C18:3, and exogenous application of C18:3 can increase FL [53,54]. Collectively, these findings provide direct genetic evidence for the co-regulation of cottonseed nutritional and fiber traits, supporting the feasibility of simultaneous improvement in breeding programs.

### 4.5. Application and Breeding Prospects of Cottonseed Nutritional Quality

In summary, cotton breeding advancements have continuously improved fiber yield and quality, accompanied by a gradual increase in cottonseed protein content. However, enhancing seed oil content has become increasingly challenging. In this study, we employed a PCA-FMF approach to screen 13 germplasms with superior comprehensive traits. These elite resources can serve as parental lines for subsequent coordinated improvement programs or as materials for mining key genes associated with cottonseed nutritional quality and fiber traits. With the constrained development of China’s soybean industry, the exploitation of cottonseed protein has garnered growing attention [27]. Cottonseed protein demonstrates broad application prospects in animal feed. Processed cottonseed protein concentrate contains over 60% protein substantially higher than commonly used cottonseed meal, enabling partial substitution of high-end proteins like soybean meal and fish meal in feed formulations [7,33]. Moreover, low-temperature pressing preserves the functional properties of cottonseed protein, enabling its use as an emulsifier in dairy alternatives and meat products [55]. Meanwhile, ongoing research into gossypol has uncovered its antiviral properties, holding promise for livestock applications [35] and potentially elevating cottonseed utilization value. Notably, cottonseed protein is generally deficient in lysine [30], making genetic engineering-mediated supplementation a promising target for future breeding programs.

## 5. Conclusions

This study assessed 26 nutritional traits of cottonseeds and 8 fiber traits across 259 upland cotton germplasms. Protein and oil emerged as the primary components, making up 76.46% of the kernel weight. Among the 16 detected amino acids, 8 are EAAs. UFAs dominated the fatty acid profile, with C18:2 being the most abundant at 55.92%. Nutritional quality varied noticeably by geographical region and breeding period. Additionally, significant positive correlations were found between nutritional traits (protein, amino acids, C18:2) and key fiber traits such as LP and FL. Thirteen elite accessions with superior combined performance were identified. These findings provide valuable data and theoretical support for high-value cottonseed utilization and the synergistic breeding of varieties with both excellent nutritional and fiber qualities.

## Figures and Tables

**Figure 1 foods-14-02895-f001:**
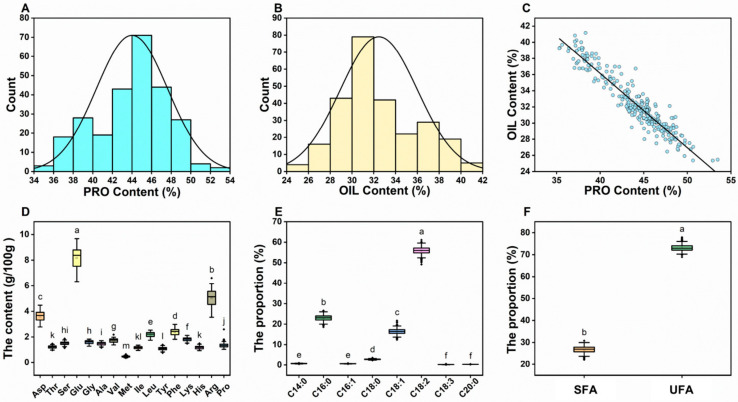
Descriptive analysis of nutritional quality of 259 upland cotton accessions. Frequency distribution of protein content (**A**) and oil content (**B**) in cottonseeds. Correlation analysis of protein content and oil content in cottonseeds (**C**). Multiple comparisons of amino acid content (**D**), fatty acid proportion (**E**), and proportion of saturated and unsaturated fatty acids (**F**) in cottonseeds. Different lowercase letters indicate significant differences at the *p* < 0.05 level. EAAs: Essential amino acids, NAAs: non-essential amino acids, Asp: Aspartic Acid, Thr: Threonine, Ser: Serine, Glu: Glutamic Acid, Gly: Glycine, Ala: Alanine, Val: Valine, Met: Methionine, Ile: Isoleucine, Leu: Leucine, Tyr: Tyrosine, Phe: Phenylalanine, Lys: Lysine, His: Histidine, Arg: Arginine, Pro: Proline, PRO: Protein, C14:0: Myristic Acid, C16:0: Palmitic Acid, C16:1: Palmitoleic Acid, C18:0: Stearic Acid, C18:1: Oleic Acid, C18:2: Linoleic Acid, C18:3: α-Linolenic Acid, C20:0: Arachidic Acid, UFA: Unsaturated fatty acids, SFAs: Saturated fatty acids.

**Figure 2 foods-14-02895-f002:**
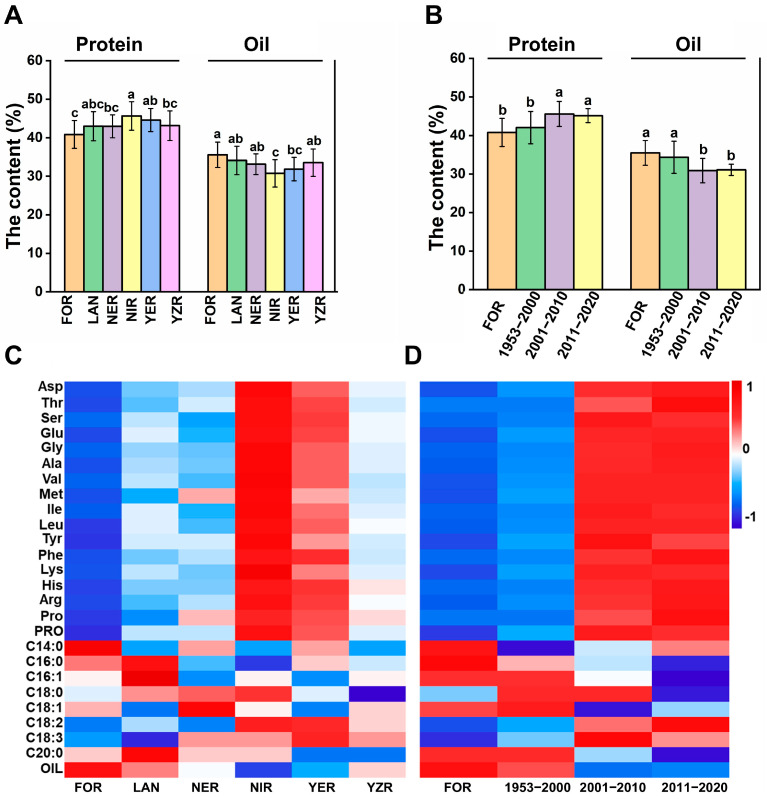
Analysis of nutritional quality differences in 259 upland cotton accessions by geographical sources and breeding period. Multiple comparisons of protein content and oil content in cottonseeds from different geographical origins (**A**) and different breeding periods (**B**). Heatmap of 26 cottonseed nutritional quality traits by geographical origins (**C**) and breeding period (**D**). Red indicates high content; blue indicates low content. Intensity reflects magnitude. Different lowercase letters indicate significant differences at the *p* < 0.05 level. YZR: Yangtze River Basin, YZR: Yellow River Basin, NER: northern extra-early-maturing cotton region, NIR: northwest inland region, LAN: landraces, FOR: foreign.

**Figure 3 foods-14-02895-f003:**
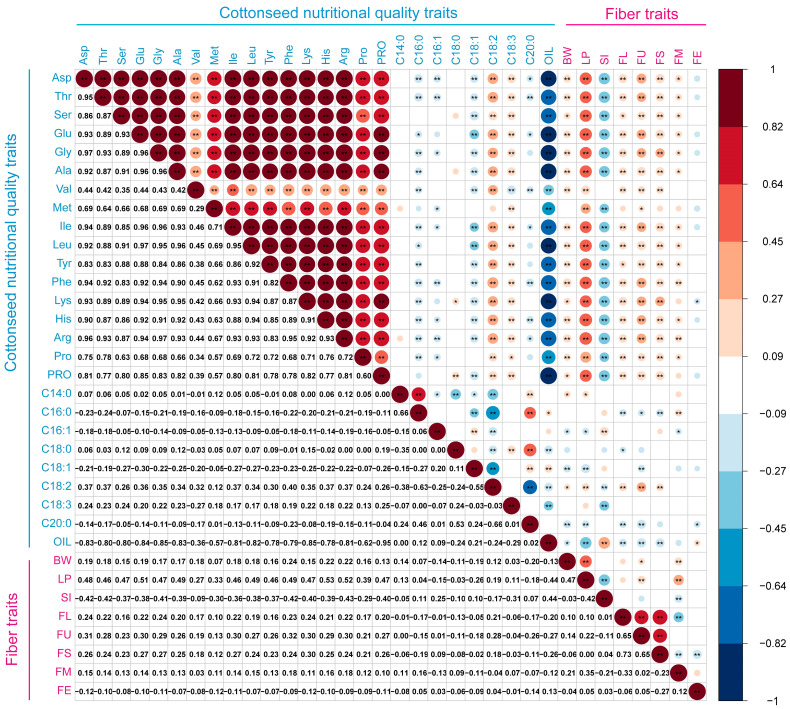
Correlation analysis of 26 nutritional quality traits and 8 fiber traits. The upper part of the heatmap shows correlation visualization (circular color blocks + significance markers), while the lower part presents the corresponding correlation coefficient values (*: *p* < 0.05, **: *p* < 0.01). Negative correlations are denoted in blue, positive correlations in red, with color intensity and circle size proportional to the strength of the correlation coefficient.

**Figure 4 foods-14-02895-f004:**
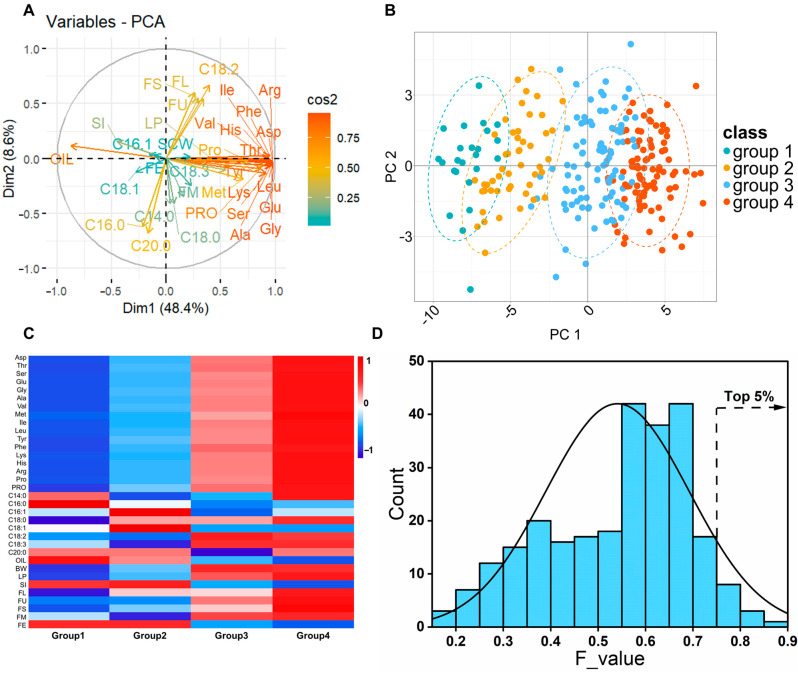
Comprehensive evaluation of 259 cotton germplasms. PCA (**A**) and cluster analysis (**B**) of 26 cottonseed nutritional quality traits and 8 fiber traits. (**C**) Comparative analysis of all traits across four groups of accessions. (**D**) Frequency distribution of comprehensive evaluation F values.

## Data Availability

Data are contained within article and Appendix A. Further inquiries can be directed to the corresponding author.

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
