# Peer review of "Comprehensive Evaluation of Nutritional Quality Diversity in Cottonseeds from 259 Upland Cotton Germplasms"

_foods, 2025, doi:10.3390/foods14162895_

Round 1
Reviewer 1 Report
Comments and Suggestions for Authors
The manuscript titled "Comprehensive Evaluation of Nutritional Quality Diversity in Cottonseeds from Upland Cotton Germplasms" is a well-structured and articulated study. However, I have some suggestions for improvement that could enhance the manuscript further.
- Figures 1–4 present crucial visual information, but their interpretation can be enhanced by incorporating: clear axis units protein content (%) instead of PRO, use of statistical indicators ( for example error bars, p-values, significance letters) and I suggest to define of grouping variables (e.g., FOR, YZR) in figure captions, not just in methods.
- The correlation matrix illustrated in Figure 3 is quite complex. It might be beneficial to: Visually group variables or separate nutritional traits from fiber traits. And to Include commentary that emphasizes biologically significant correlations (such as LP with amino acids and C18:2 relating to fiber strength).
- In the sections on amino acid and fatty acid analyses, there should be more clarity regarding: the duplicate measurements and quality control samples on the criteria for data acceptance, if it's possible, the use of standards and metrics for validation.
- Terms like PRO, BW, and LP are inconsistently formatted, sometimes appearing in uppercase and other times in mixed case. Maintaining standard terminology throughout the manuscript, consistently using “Protein,” “Boll Weight,” and “Lint Percentage,” will improve overall readability.
- The concurrent enhancement of fiber and seed quality is a fascinating topic. It would be beneficial to explore: Potential explanations related to linkage or pleiotropy, and also Instances of breeding lines or gene clusters that have demonstrated this convergence.
- Several key citations [1], [21], [39] play an essential role in the arguments put forth in the Discussion section. It would be useful to highlight their relevance with brief descriptors (for example as demonstrated by Wang et al. [21], gossypol has antiviral properties.
- The application aspect can be strengthened by briefly mentioning how elite germplasms (like Xinluzao29) can be incorporated into breeding programs or trials.
Author Response
Comments 1: Figures 1–4 present crucial visual information, but their interpretation can be enhanced by incorporating: clear axis units protein content (%) instead of PRO, use of statistical indicators (for example error bars, p-values, significance letters) and I suggest to define of grouping variables (e.g., FOR, YZR) in figure captions, not just in methods.
Response 1: Thank you for your constructive feedback. We have addressed your suggestions as follows: In Figures 1A and 1C, "PRO" has been revised to "protein content (%)" to explicitly indicate the axis units, enhancing readability. For Figure 2, the grouping variables "FOR", "YZR" etc are now defined in the figure caption (in addition to their mention in the Methods section) to ensure immediate clarity for readers. After carefully reviewing all figures, we have added statistical indicators to Figures 1D-F, 2A-B, and Figure 3. There is no need to include significance indicators in the figures.
Comments 2: The correlation matrix illustrated in Figure 3 is quite complex. It might be beneficial to: Visually group variables or separate nutritional traits from fiber traits. And to Include commentary that emphasizes biologically significant correlations (such as LP with amino acids and C18:2 relating to fiber strength).
Response 2: Thank you for your insightful suggestions regarding Figure 3. We have revised the correlation matrix as follows: Visual grouping of variables has been implemented in Figure 3, with cottonseed nutritional quality traits and fiber quality traits now distinguished by distinct colors in both the x-axis and y-axis labels, enhancing the clarity of their separation. Additionally, we have expanded the commentary in Section 3.4 of the text to emphasize the biological significance of key correlations.
Comments 3: In the sections on amino acid and fatty acid analyses, there should be more clarity regarding: the duplicate measurements and quality control samples on the criteria for data acceptance, if it's possible, the use of standards and metrics for validation.
Response 3: Thank you for your valuable suggestion. We have revised the amino acid and fatty acid analyses in paragraph 2.4 (line 163) and paragraph 2.5 (line 183) of the revised manuscript to clarify the details of duplicate measurements, quality control criteria, and validation metrics.
Comments 4: Terms like PRO, BW, and LP are inconsistently formatted, sometimes appearing in uppercase and other times in mixed case. Maintaining standard terminology throughout the manuscript, consistently using “Protein,” “Boll Weight,” and “Lint Percentage,” will improve overall readability.
Response 4: Thank you for pointing out the inconsistency in terminology formatting. We have revised the manuscript to standardize these terms: "PRO" has been uniformly replaced with "protein" throughout the text to ensure clarity. For "BW" (Boll Weight) and "LP" (Lint Percentage), we have retained their abbreviations for conciseness after their first occurrence, where they are explicitly defined with their full names. This approach balances readability with brevity, allowing smooth flow while ensuring readers unfamiliar with the terms can easily reference their definitions upon first encounter.
Comments 5: The concurrent enhancement of fiber and seed quality is a fascinating topic. It would be beneficial to explore: Potential explanations related to linkage or pleiotropy, and also Instances of breeding lines or gene clusters that have demonstrated this convergence.
Response 5: Thank you for highlighting this important direction for expanding our discussion. To address your suggestion, we have added content in Section 4.4 (line 470 and 485) to explore potential genetic mechanisms underlying the concurrent improvement of fiber and seed quality, as well as relevant empirical evidence.
Comments 6: Several key citations [1], [21], [39] play an essential role in the arguments put forth in the Discussion section. It would be useful to highlight their relevance with brief descriptors (for example as demonstrated by Wang et al. [21], gossypol has antiviral properties.
Response 6: Thank you for your helpful suggestion. We have revised the presentation of citations [21] (now [34] in the revised manuscript) and [39] (now [51] in the revised manuscript) in Sections 4.1 (line 397) and 4.4 (line 479) to clarify their specific relevance to the arguments. Regarding citation [1], as it is a review article that provides a broad overview of historical cotton breeding priorities rather than specific experimental findings, we have retained the general reference format.
Comments 7: Thank you for your valuable suggestion. We have added relevant content in Section 4.5 to address this point.
Response 7: Thank you for your valuable suggestion. We have added relevant content in Section 4.5 (line 492) to address this point.
Reviewer 2 Report
Comments and Suggestions for Authors
Dear Authors,
Some general questions about the work remain, especially considering the industrial processing and uses of cottonseed meal and bran as high value-added protein and oil products. What is “deep processing”?
The cost of the process, feasibility, and availability of products should be compared with soybean processing and by-products. A poor comparison is made considering conventional or similar processes/products.
What do the measured fiber quality parameters mean? Why are they important? What are the acceptable values for them?
The resolution of the figures should be improved.
Additional suggestions, requests, and corrections are presented as comments in the PDF manuscript.

Author Response
Comments 1: Some general questions about the work remain, especially considering the industrial processing and uses of cottonseed meal and bran as high value-added protein and oil products. What is “deep processing”?
Response 1: Thank you very much for your question. The so-called deep processing of cottonseed mainly involves two directions: Cottonseed protein preparation: Through processes such as deoiling, removal of anti-nutritional factors (e.g., gossypol), and fermentation, cottonseed meal is processed into high-quality protein concentrates for feed applications, which can partially replace soybean meal and fish meal. Cottonseed oil processing: It employs advanced extraction technologies (microwave-assisted, supercritical COâ‚‚) to enhance efficiency and quality, followed by dephenolization and refining (degumming, neutralization, bleaching, deodorization) to remove impurities like gossypol. Modification via hydrogenation or interesterification optimizes fatty acids, enabling high-value use in food, pharmaceuticals, and biofuels.
Comments 2: The cost of the process, feasibility, and availability of products should be compared with soybean processing and by-products. A poor comparison is made considering conventional or similar processes/products.
Response 2: Thank you very much for your valuable suggestion. We fully agree with your insight. As this study primarily focuses on investigating the variation in cottonseed nutritional quality at the early stage, we have not delved deeply into the subsequent processing and utilization of cottonseeds. However, we plan to address the issues you raised in our follow-up experiments. Here, we would like to briefly discuss your question: while specific cost details and market availability data are not elaborated in the current manuscript, the mature de-phenolization and processing technologies (e.g., low-temperature extraction) for cottonseed protein concentrate [7,8] have laid the foundation for cost control in industrial production. Additionally, the widespread cultivation of upland cotton globally ensures a stable raw material supply, which is conducive to ensuring product availability. Further research on economic feasibility and market distribution could be a valuable direction for future studies to complement the application potential of cottonseed deep processing.
Comments 3: What do the measured fiber quality parameters mean? Why are they important? What are the acceptable values for them?
Response 3: Thank you for your question. As the primary product of cotton production, cotton fiber quality directly dictates the economic value of cotton in the textile and related industries. In breeding practices, enhancing fiber yield and quality traits remains the core objective; thus, improving cottonseed nutritional quality should not come at the expense of these critical fiber traits. To explore the feasibility of their coordinated improvement, we measured 8 fiber traits and analyzed their correlations with cottonseed nutritional traits. As highlighted in our study, the synergistic enhancement of both fiber quality and seed nutritional value is vital for maximizing the overall economic benefits of cotton production. Regarding the acceptable ranges for these fiber traits, there are no rigid universal standards due to variations in growing environments and varieties. However, general breeding goals prioritize higher LP, longer FL, and greater FS, as these characteristics are consistently associated with enhanced textile processing performance and market value.
Comments 4: The resolution of the figures should be improved.
Response 4: Thank you very much for your suggestion. We have revised the figure 2 in the manuscript to improve their resolution as recommended, and all original figures with enhanced resolution have been uploaded.
The following are the questions marked by the reviewer in the original manuscript.
Comments 5: In line 109 of the manuscript, mention the reference methodologies in the text.
Response 5: Thank you very much for your suggestion. We have revised the figure 2 in the manuscript to improve their resolution as recommended, and all original figures with enhanced resolution have been uploaded.
Comments 6: In paragraph 3.1 of the manuscript, please clarify the percentage of protein, oil, and amino acid contents relative to cottonseed weight—whether the cottonseed weight is based on wet weight or dry weight.
Response 6: Thank you for your suggestion. All the percentages of protein, oil, and amino acid contents are based on dry weight of cottonseeds.
Comments 7: In paragraph 3.1 of the original manuscript, you mentioned: "Lys, a common limiting amino acid in plant-based feeds, has a content range of 1.48% to 2.12%. The coefficient of variation (CV) for the 16 amino acids ranges from 7.79% to 13.07%, with Asp, Tyr, Phe, His, Arg, and Pro showing CVs exceeding 10%." "The CV for the 8 fatty acids ranges from 3.42% to 26.37%, with C18:2 and C14:0 showing higher CVs of 26.37% and 13.74%, respectively." "All traits exhibited normal distributions. CVs ranged from 1.46% (FU) to 13.34% (BW); four traits had CV > 10%. Yield traits showed greater variation (all CV > 10%) than quality traits, with BW ranging from 71.41 g to 163.63 g. Among quality traits, only FM had CV > 10% (Table S5). What does that mean? Is it good?
Response 7: Thank you for your question. The purpose of presenting these coefficients of variation (CV) results is to indicate that there is substantial genetic variation in amino acid contents, fatty acid compositions, and fiber quality traits within the natural population of upland cotton germplasms. Such a high degree of variation is beneficial as it provides a broad genetic basis for targeted high-value utilization of cottonseeds (e.g., breeding varieties with enhanced specific amino acids or fatty acids) and the screening of elite germplasms with superior nutritional and fiber quality traits, which is valuable for subsequent breeding programs and industrial applications.
Comments 8: In paragraph 3.2, regarding the "Analysis of Differences in Nutritional Quality of Cottonseeds among Different Geographical Sources and Breeding Periods", what are the plausible explanations for these behaviors? For example, soil nutrition, availability of rain or irrigation, climate, etc.
Response 8: Thank you for your question. We have addressed this in paragraph 4.3 of the Discussion, where we note that this shift in nutritional quality across geographical sources and breeding periods may relate to changing cultivation patterns, potentially involving fatty acid degradation for water production under drought stress in arid regions. This aligns with the ecological adaptation of cotton varieties to diverse environmental conditions (e.g., arid vs. humid regions) and long-term artificial selection pressures, which collectively contribute to the observed variations in nutritional traits.
Comments 9: In paragraph 3.2, you mentioned: "However, these accessions had lower protein content, total amino acid content and C18:2 proportion. NIR accessions had the highest protein (45.65%) and all 16 amino acids. Although their oil content was lowest (30.75%), their C18:2 proportion was highest (56.34%). Overall, the NIR and YER accessions exhibited higher protein, amino acid content and C18:2 proportion, while FOR, LAN, YZR, and NER accessions had higher oil content (Figure 2C), which is a noteworthy finding." Why is this finding noteworthy?
Response 9: Thank you for your question. This finding is considered noteworthy for the following reasons: In different cotton-growing regions, NIR and YER accessions exhibit higher protein and amino acid contents but lower oil content, which aligns with the significant negative correlation between protein and oil content observed in our study (Figure 1C). However, unexpectedly, these accessions also show a higher proportion of C18:2 (a major unsaturated fatty acid accounting for 55.92% of total fatty acids). Normally, given their lower total oil content, a lower proportion of C18:2 might be anticipated. This counterintuitive result is valuable because C18:2 is a key unsaturated fatty acid with well-documented health benefits, such as antioxidant properties and contributions to cardiovascular health [10,30]. A higher C18:2 proportion in NIR and YER accessions enhances the nutritional quality of cottonseed oil, even with reduced total oil content. This unique combination of high protein, high C18:2, and low oil content in these regions provides a distinct germplasm resource for breeding varieties tailored to dual-purpose applications (e.g., high-quality protein feed and functional oil production), which is of practical significance for the high-value utilization of cottonseeds.
Comments 10: In paragraph 3.3, you stated: "All traits exhibited normal distributions. CVs ranged from 1.46% (FU) to 13.34% (BW); four traits had CV > 10%. Yield traits showed greater variation (all CV > 10%) than quality traits, with BW ranging from 71.41 g to 163.63 g. Among quality traits, only FM had CV > 10% (Table S5). What does that mean? Is it good?
Response 10: Thank you for your question. We consider the observed variation in fiber traits to be favorable. The presence of substantial variation, particularly the higher coefficient of variation in yield traits (all CV > 10%) and the noticeable variation in fiber micronaire (FM) among quality traits, is meaningful for our research. Such differences provide a solid foundation for subsequent variety improvement. If there were no variation in these traits, it would limit the potential for selecting superior germplasms or conducting targeted breeding to enhance fiber yield and quality, thereby reducing the practical value of the study. Thus, the observed variation is crucial for advancing cotton breeding and improving the economic value of cotton varieties.
Comments 11: In paragraph 3.4, regarding the "Correlation Analysis of Cottonseed Nutritional Quality and Fiber Traits", do you have any explanation for this behavior?
Response 11: Thank you for your question. Fibers and cottonseeds are the two main products of the cotton production process, and the concurrent variation between their traits is a notable phenomenon. This observed correlation is likely driven by genes with pleiotropic effects or linked genetic loci that co-regulate both nutritional quality (e.g., protein, amino acids, fatty acids) and fiber traits (e.g., lint percentage, fiber length). For instance, previous studies have indicated that genes involved in carbon allocation and lipid metabolism (such as fatty acid desaturases) can simultaneously influence seed oil synthesis and fiber development [51, 53], supporting the idea that shared genetic regulatory networks underlie these traits. This linkage provides a potential genetic basis for the coordinated improvement of both nutritional and fiber qualities in cotton breeding.
Comments 12: In figure 3 Regarding the description "Lower triangle: Pearson correlation coefficients; upper triangle: significance levels (*: p < 0.05, **: p < 0.01). Negative...", there may be ambiguity.
Response 12: Thank you for your suggestion. We have revised this sentence in line 329 of the manuscript to clarify the description of the correlation heatmap.
Comments 13: Why didn't you integrate the Discussion and Results sections?
Response 13: Thank you for your suggestion. We appreciate the insight, but when preparing the manuscript, we referred to the writing guidelines of this journal and its recently published articles, which typically follow a structure that separates the Results and Discussion sections. We hope this structure meets the journal's requirements.
Comments 14: In paragraph 4.1, you mentioned cottonseed-related by-products. To clarify: Cottonseed meal and cottonseed bran are by-products of kernel oil extraction, correct? Can cottonseed meal and bran be used as animal feed? Do they contain anti-nutritional compounds? Is industrial processing the same as for soybean meal and soybean bran?
Response 14: Thank you for your question. Cottonseed meal and cottonseed bran are indeed main by-products of cottonseed kernel oil extraction, similar to soybean meal and bran derived from soybean oil processing. Cottonseed meal and bran can be used as animal feed, especially after proper processing. As noted in our manuscript, cottonseed protein concentrate (produced from cottonseed meal via de-phenolization and purification) can effectively replace fish meal and soybean protein in feed formulations [7,8]. Raw cottonseed meal and bran contain anti-nutritional compounds, with gossypol being the most notable toxic substance, which historically limited their application in feed [32]. However, modern de-phenolization technologies (e.g., solvent extraction, enzymatic treatment) are mature and can effectively remove excessive gossypol to safe levels [28,33]. Notably, recent studies (e.g., Wang et al. [34]) have revealed that low concentrations of gossypol exhibit broad-spectrum antiviral activity in livestock, adding value to its controlled use.
Comments 15: In paragraph 4.1, regarding cottonseed deep processing, what about the cost of the process? What about product availability?
Response 15: Thank you for your suggestion. As addressed in our response to your second question, we have already provided relevant explanations on this matter. Please refer to that response for details.